# Depolarization of mouse DRG neurons by GABA does not translate into acute pain or hyperalgesia in healthy human volunteers

Kyra Sohns[ORCID][�], Anna Kostenko[�], Marc Behrendt, Martin Schmelz[ORCID]*, Roman Rukwied, Richard Carr

Experimental Pain Research, MCTN, Medical Faculty Mannheim, Heidelberg University, Heidelberg, Germany

euro These authors contributed equally to this work.
* martin.schmelz@medma.uni-heidelberg.de

**Data Availability Statement:** All relevant data are within the manuscript and its Supporting Information files.

## Abstract

The majority of somatosensory DRG neurons express $GABA_A$ receptors ($GABA_AR$) and depolarise in response to its activation based on the high intracellular chloride concentration maintained by the Na-K-Cl cotransporter type 1 (NKCC1). The translation of this response to peripheral nerve terminals in people is so far unclear. We show here that GABA ($EC_{50}$ = 16.67µM) acting via $GABA_AR$ produces an influx of extracellular calcium in approximately 20% (336/1720) of isolated mouse DRG neurons. In contrast, upon injection into forearm skin of healthy volunteers GABA (1mM, 100µl) did not induce any overt sensations nor a specific flare response and did not sensitize C-nociceptors to slow depolarizing electrical sinusoidal stimuli. Block of the inward chloride transporter NKCC1 by furosemide (1mg/100µl) did not reduce electrically evoked pain ratings nor did repetitive GABA stimulation in combination with an inhibited NKCC1 driven chloride replenishment by furosemide. Finally, we generated a sustained period of C-fiber firing by iontophoretically delivering codeine or histamine to induce tonic itch. Neither the intensity nor the duration of histamine or codeine itch was affected by prior injection of furosemide. We conclude that although GABA can evoke calcium transients in a proportion of isolated mouse DRG neurons, it does not induce or modify pain or itch ratings in healthy human skin even when chloride gradients are altered by inhibition of the sodium coupled NKCC1 transporter.

## Introduction

$GABA_AR$ are chloride and bicarbonate permeable anion channels. The direction of ion flux through $GABA_A$ is determined by the intracellular chloride concentration that is regulated by members of the Slc12 family of electroneutral solute carriers. In neurons, the sodium coupled NKCC1 transporter mediates inward chloride transport while outward KCC chloride transporters are coupled to potassium. In most neurons, KCC2 predominates and maintains a low intracellular chloride concentration such that activation of $GABA_AR$ results in chloride influx and neuronal hyperpolarization. However, in some neurons, including dorsal root ganglion

**Funding:** The study was supported by the German Research Foundation (Deutsche Forschungsgemeinschaft, https://www.dfg.de/en/research-funding/funding-opportunities), grant project 397846571 (RR) and FOR 2690 (MS, RR) and SFB 1158 (RC, MS). The funders had no role in study design, data collection and analysis, decision to publish, or preparation of the manuscript.

**Competing interests:** The authors have declared that no competing interests exist.

**Abbreviations:** DMEM, Dulbecco's Modified Eagle's Medium; DMSO, Di-Methyl-Sulfoxid; DPBS, Dulbecco´s Phosphate Buffered Saline; DRG, Dorsal Root GanglionFBS: Fetal Bovine Serum; ECK, ElectroCardioGram; GABA, Gamma-Amino-Butyric Acid; GABA$_A$R, Gamma-Amino-Butyric Acid Receptor Type A; HBBS, Hanks' Balanced Salt Solution; HEPES, -2-Hydroxyethylpiperazine-N-2-Ethane Sulfonic Acid; KCC2, Potassium (K)-Chloride (C)-Cotransporter (C) Type 2; LDI, Laser Doppler Imaging; NKCC1, Sodium (N)-Potassium (K)-Chloride (C)-Cotransporter (C) Type 1; NKCC2, Sodium (N)-Potassium (K)-Chloride (C)- Cotransporter (C) Type 2; NRS, Numerical Rating Scale; OSR1, Odd-Skipped Related Transcription Factor 1; PBS, Phosphate Buffered Saline; ROI, Region of Interest; SPAK, Ste20-related Proline-Alanine-rich Kinase; TTA-A2, (2-(4-cyclopropylphenyl)-*N*-((1*R*)-1-(5-((2,2,2-trifluoroethyl)oxo)-pyridin-2-yl)ethyl) acetamide; TTX, Tetrodotoxin.

(DRG) neurons [1, 2], trigeminal ganglion neurons [3], hypothalamic posterior pituitary neurons [4], spiral ganglion neurons [5] and sympathetic post-ganglionic neurons [6] NKCC1 expression prevails to maintain an elevated intracellular chloride resulting in chloride outflow and thus depolarizing currents in response to GABA$_A$R activation. For DRG neurons, depolarizing responses to GABA are evident in all neuronal compartments including the cell body, peripheral axons [7] and central projections within the dorsal root. Depolarizing GABA responses are based on the NKCC1-generated chloride gradient [8] and can manifest as a transient increase in intracellular calcium in isolated DRG neurons [3]. While increasing axonal excitability to repetitive electrical stimulation [7] the depolarizing effect of GABA can reduce neuronal excitability via inactivation of voltage-gated sodium channels and open chloride channels will shunt generator potentials and inhibit action potential generation [9]. Accordingly, the term "double-edged sword" [10] has been used for the opposing functional consequences, presynaptic inhibition and dorsal root reflexes.

In addition to the regulation of intracellular chloride concentration by inward (NKCC1) and outward (KCC2) chloride transport during neuronal development [11] changes in intracellular chloride are also observed with changes in cell volume [12]. Additionally, in neurons, axotomy is accompanied by increased intracellular chloride concentrations that can be regulated dynamically in association with pathology [13] or in response to overt injury [14]. In response to skin injection of inflammatory stimuli, neurons in the spinal dorsal horn downregulate KCC2 expression [15] leading to an increase in their intracellular chloride concentration and a subsequent reduction in the efficacy of post-synaptic inhibitory transmission. Similarly, primary afferent neurons respond to inflammatory stimuli with an increase in intracellular chloride [16] that occurs within hours along with enhanced NKCC1-phosphorylation [17]. In particular, the increase in the reversal potential of GABA$_A$R mediated currents in DRG neurons [16] arises from increased NKCC1 activity through SPAK and OSR1 mediated phosphorylation [18], however with little to no change in NKCC1 expression [19]. Interestingly, the functional class of DRG neurons in which chloride is upregulated differs depending upon the type of injury [16].

Changes in neuronal intracellular chloride have been implicated in the progression of a number of human pathologies including above-mentioned inflammatory nociceptive pain– the inhibition of NKCC1 via bumetanide leads to a reduction in nocifensive capsaicin-evoked behaviour [20]–and also neuropathic pain [13]–and via furosemide, to lower pruriception in chronic kidney failure [21].

A major source of chronic pain and itch is long-lasting ongoing activity of primary nociceptors [22, 23]. Computational models suggested that such ongoing activity in C-nociceptors primarily decreased their excitability via hyperpolarization, sodium channel inactivation and intracellular sodium accumulation [24]. Importantly, superfusion of isolated nerves *in vitro* with GABA has shown the opposite effect, i.e. GABA-induced hyperexcitability was more pronounced following higher discharge rates of C-fibers [7]. We therefore used intradermal injections of GABA with/without inhibition of NKCC1 in humans *in vivo* to investigate the general role of intra-axonal chloride in supporting sustained firing in sensory axons. The pruritogens codeine and histamine were used to generate a sustained period of afferent C-fiber activity. Slow depolarizing electrical stimuli with suprathreshold intensity were applied to provoke single bursts of activity, but also to induce sustained discharge in C-nociceptors, the latter of which we hypothesized to be sensitized by GABA. In contrast to results from mouse experiments, intradermal GABA or blocking NKCC1 by local furosemide alone or in combination, had no influence on itch or pain ratings in healthy volunteers.

## Materials and methods

### Ethics approval

**Mice.** Approval for harvesting dorsal root ganglia from euthanised mice was obtained in compliance with guidelines for the welfare of experimental animals as stipulated by the Federal Republic of Germany under approval number I-19/15 and I-21/15 within the Medical Faculty Mannheim of Heidelberg University.

**Human subjects.** Experimental procedures carried out on people were approved by the Ethics Committee II of Heidelberg University, Medical Faculty Mannheim (approval number 2016-568N-MA). Healthy volunteers aged between 22 and 58 years were recruited for the study. Exclusion criteria comprised prevailing neurological disorders, dermatological disease and use of medication to treat these within the month prior to testing. Subjects were informed about the procedure and signed a consent form before participating in the experiments. Participating subjects were seated comfortably in a chair in a quiet room with an ambient air temperature of $20\pm2°C$. The subject's forearms were positioned comfortably on a table alongside one another and nestled into an air cushion, evacuated to maintain a supportive shape.

**Calcium imaging.** Adult C57BL/6N mice (3 male, 20 female) were anesthetized with 2% isoflurane and killed by decapitation immediately prior to harvesting DRG neurons from cervical, thoracic and lumbar segments. Whole DRG explants were placed into chilled sterile HBSS ($Ca^{2+}/Mg^{2+}$ added; Thermo Fischer, Karlsruhe, Germany) before being cleaned of excess tissue and digested in HBSS with liberase (1.8U/ml, Roche, Mannheim, Germany) for 40 minutes at 37°C. Digested ganglia were triturated and transferred to DMEM with 10% FBS and 1X penicillin-streptomycin (Thermo Fischer). Cells were centrifuged for 2 minutes at 1200rpm, triturated, resuspended and centrifuged a second time before being plated onto 25mm glass coverslips housed in 35mm petri dishes and left for 90–120 minutes to adhere after which time 2ml of DMEM was added. Prior to plating, coverslips resident in their petri dishes were coated with a drop of poly-L-lysine/laminin (1:100, both from Sigma Aldrich, St. Louis, MO, USA) and incubated at 37°C for 30–90 minutes. Coverslips were then rinsed with ddH20 and a 10μl drop of laminin/ddH20 (1:10) placed on the coverslip and left to dry until plating. Cells were allowed to adhere overnight in a humidified incubator at 37°C and 5% $CO_2$.

Fluorescence calcium imaging was performed using cultured DRG neurons within 36 hours of plating. Cells were loaded with the intensity-based calcium indicator Fluo8-AM (3μM, AAT Bioquest, Sunnyvale, CA, USA) by incubation for 20–30 minutes in the dark at room temperature. Cells were then washed twice with imaging solution (see *Solutions*). The coverslip on which the cells were plated was transferred to a stage-mounted slotted bath (RC-21 BRFS, Warner Instruments, Holliston, MA, USA). Coverslips were sealed to the base of the slotted bath with an annulus of silicone grease generating a total bath volume of 263μl. The slotted bath was equipped with a pair of platinum wire electrodes to deliver electrical field stimulation with rectangular pulses (see below). The resulting imaging chamber was perfused continuously using a gravity-driven system. Chemical stimulation was delivered by hand-switching between multiple reservoirs using a manifold tap system.

Images were acquired using a back-illuminated 512x512 pixel cooled EMCCD camera (Evolve 512, Photometrics, Tucson, AZ, USA) connected to the side port of an Axiovert 200 microscope (Zeiss, Jena, Germany) imaged with a 10x objective (Neofluar) and a filter set comprising excitation BP 450–490 nm, dichroic 510 nm and emission LP 515 nm (Chroma Technologies, Bellows Falls, VT, USA). The sequence for image collection comprised an initial brightfield image and a single fluorescence image, used to subsequently check for movement and demarcate cell boundaries. Fluorescence images were acquired every 2 seconds (0.5Hz)

using a 10ms exposure to 465nm LED (Prior Scientific Instruments, Cambridge, UK). The relative timing of camera frame grabs, LED exposure and electrical stimulation were controlled using μManager software synchronised with an Arduino Duemilanove. Constant current electrical field stimulation was delivered as 1ms rectangular pulses at 10Hz (Digitimer DS7A, Welwyn Garden City, UK).

**Analysis of fluorescence images.** Image analysis was performed using ImageJ [25]. Initially, regions of interest (ROI) corresponding to individual cells were established by manually positioning circular ROIs on those cells. In addition, five background ROIs, that did not contain cells, were demarcated. The mean gray scale value determined from all five background ROIs for each image was subtracted from the fluorescence intensity values (F) of all cell-based ROIs. For cell-based ROIs, only those ROIs responding to 45mM potassium, applied at the end of each experiment, were considered for further analysis. For cell-based ROIs, fluorescence values over time were low-pass filtered using a 5 point box-car average. Fluorescence values for each cell-based ROI were then normalized. Normalized fluorescence ($F/F_0$) traces were calculated for each ROI by dividing fluorescence values by an average initial fluorescence ($F_0$) determined as the average fluorescence in that ROI during the first 20 images.

To determine whether cells responded to a given stimulus a simple threshold crossing assessment was performed. A threshold was defined for each cell and each stimulus by averaging $F/F_0$ values over the 10 images immediately preceding stimulation and then denoting threshold as this mean plus three standard deviations. A cell-based ROI was considered to have responded to a stimulus if the mean $F/F_0$ value during stimulation was greater than the threshold value. Response traces for all cells were inspected individually in addition to thresholding analyses. In general, cells were excluded from analysis if, after a stimulus-induced an increase in fluorescence the $F/F_0$ value had not returned within 3 minutes to less than 10% increase above the initial fluorescence, i.e. immediately preceding the stimulus (baseline). Additionally, cells that exhibited calcium transients in the absence of a stimulus were excluded. For concentration-response data, calcium responses for each concentration were normalized by division of the response with the average calcium transient to two 30 second 200μM GABA applications in a sequence. The TTX-ratio of DRG neurons responding to 10Hz stimulation (Fig 1N) was calculated as a quotient of the calcium signal upon stimulation under control conditions and with TTX.

## Psychophysical assessment of itch and pain

Itch and pain intensity were quantified by each subject using an analogue 11-point numerical rating scale (NRS) that spanned values from 0–10. Subjects were told that the scale was anchored at zero to indicate "no itch" or "no pain" and at ten as the "most intense imaginable itch" or "maximum imaginable pain". Subjects were asked to rate sensations of itch or pain using this NRS scale and additionally to categorize the quality of the itch or pain sensation into categories of burning, stabbing, stinging or pricking. Subjects were also encouraged to report any additional qualifying sensations with descriptors such as prickle, pins and needles or tingle.

## Histamine and codeine to induce itch

Experimental itch was evoked by iontophoresis of either 1% (w/v in de-ionized water, Merck Millipore) histamine dihydrochloride or codeine phosphate hemihydrate. For iontophoresis, constant current was applied at 1mA for 20 seconds (charge 20mC) using a battery powered stimulator (Perilont PF 382b, Perimed AB, Stockholm, Sweden). A piece of soft porous paper 3–4 mm in diameter was soaked with histamine or codeine solution and placed beneath a

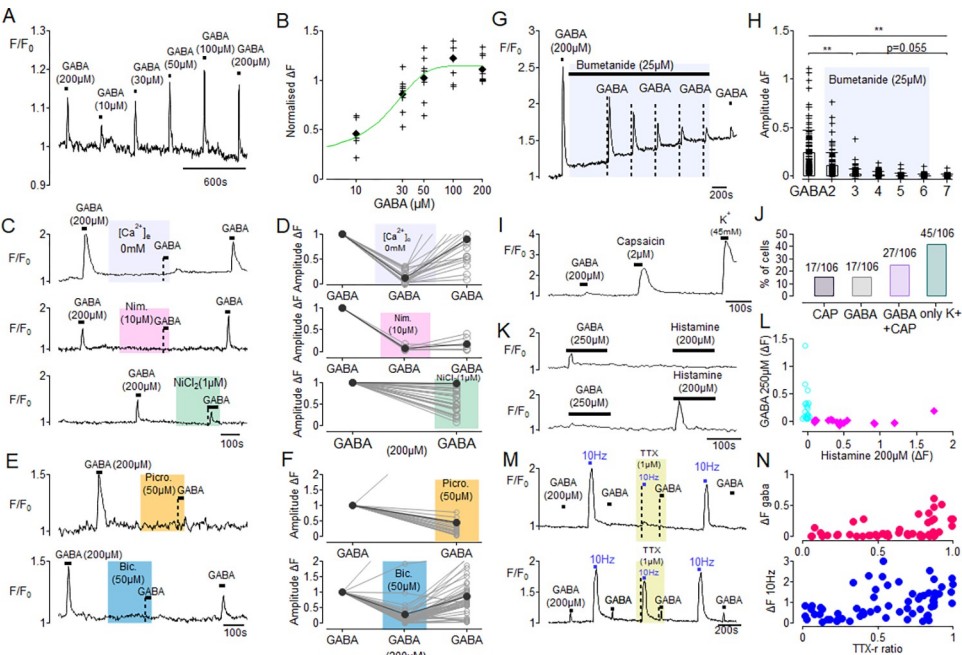

**Fig 1. Pharmacological activation of GABA_A elicits calcium transients in isolated mouse DRG neurons.** A) Bath applied GABA (10–200μM) elicits a concentration-dependent increase in intracellular calcium in DRG neurons (n = 8, 2 dishes). B) Sigmoidal plot depicting the concentration-response curve of GABA from 10–200μM ($EC_{50}$ = 16.67μM). C) Absence of extracellular calcium (0 mM $Ca^{2+}_e$, upper trace, n = 20, 5 dishes) abolished the calcium response to GABA, as did nimodipine (10μM, centre trace, n = 6, 1 dish). Nickel (1μM, NiCl2, lower trace) produced partial block in n = 47 (4 dishes). D) Amplitude values normalised to the initial GABA response depict that the absence of extracellular calcium mediates the GABA response (RM-ANOVA, p<0.001), which returns upon wash-in of standard imaging buffer. Nimodipine abolishes the calcium transient (Friedman's test, p<0.001), which remains reduced upon washout, while GABA responses in NiCl2 remain almost the same (paired t-test, p<0.001). E) Representative traces of GABA responses with block by the allosteric GABA_AR antagonist picrotoxin (50μM, upper trace) and the competitive antagonist bicuculline (50μM, lower trace) show that GABA responses were mediated by GABA_AR. F) Normalised responses to GABA (200μM) in presence of picrotoxin (paired t-test, p = 0.026, n = 19, 3 dishes) and bicuculline (RM-ANOVA, p<0.001, n = 47, 8 dishes) show a reduction of GABA-evoked calcium transients, which in case of bicuculline return upon washout. G) Raw trace depicting normalized calcium transients of DRG neurons in response to repetitive GABA (200μM) stimulation during incubation with bumetanide (25μM, black bar and light purple shading). H) Pooled amplitude values of all neurons (n = 122, 6 dishes) showing the progressive decline in response to repetitive applications of GABA in presence of bumetanide (25μM, light purple shading) (RM-ANOVA, p<0.001), with no difference between the GABA 1 and GABA 2 application (n.s.). A significant reduction between GABA 1 and GABA 3 was apparent, but no change in amplitude between GABA 3 –GABA 7 (n.s.), indicating absent GABA responses upon depletion of the chloride gradient. I) Raw trace depicting a neuron responding to both GABA (200μM) & Capsaicin (2μM) and response to high K+ solution (45mM). J) The number of cells expressed as percentage of total cells recorded (n = 106, 3 dishes), responding to only capsaicin (2μM, left column), or GABA (200μM), to GABA (200μM) + capsaicin (2μM), and to high $K^+$ solution only (right column). K) Of 27 (approx. 18%, 2 dishes) DRG neurons that responded with a calcium signal to GABA (250μm) none showed a calcium response to histamine (200μM, upper trace) and vice versa for histamine (approx. 9%, lower trace). L) Scatterplot depicting amplitudes of GABA responders (cyan) plotted against histamine responders (magenta) to demonstrate the result of no overlapping DRG cells responding to both stimuli. M) TTX-S electrically-responsive cells are less GABA (200μM) responsive (upper trace) and TTX-R electrically-responsive cells are more likely to show a GABA response (lower trace). N) The amplitude of calcium responses to electrical stimulation showed a composition of both TTX-sensitive and TTX-resistant components, as indicated by TTX-R ratios between 0.5 and 1. (upper trace in M). The amplitude of calcium responses to GABA tended to be larger in cells with a lower TTX sensitivity index, i.e. a higher TTX-R component (upper trace M, shows no responses to 10Hz stimulation and GABA during TTX, while lower trace M, reveals responses).

bespoke silver-silver chloride electrode of 5mm diameter that served as the anode (+). The anode was positioned at the skin site at the centre of the region scanned by laser Doppler imaging (see below) either 10 cm distal to the cubital fossa or 10 cm proximal to the wrist. A

disposable infant ECG electrode (Covidien Medtronic Deutschland GmbH, Neustadt a.d. Donau, Germany) served as the cathode (-) and was positioned at the wrist. Subjects were asked to rate itch on the NRS immediately after iontophoresis and then at 30 second intervals for 10 minutes.

## Electrically induced pain

Pain ratings were determined in response to sinusoidal electrical current stimuli delivered either as a single 1Hz half period sinusoidal pulse (duration 500ms, amplitudes 0.2, 0.6 or 1mA) or as a continuous 4Hz stimulus (duration 60 s, amplitude 0.2mA). Constant current stimuli (DS5, Digitimer Ltd., Welwyn Garden City, U.K.) were delivered as sinusoidal profiles via a Digital-Analogue Converter (DAQ) (NI USB-6221, National Instruments, Texas, US) with the timing controlled by dedicated software (Dapsys 8.0, Brian Turnquist, Minnesota, US). Electrical stimuli were applied transcutaneously via a pair of parallel platinum wire electrodes separated by 2 mm and each 0.4 mm in diameter (Nørresundby, Denmark). The electrodes were mounted in 3D-printed plastic and held on the subject's skin by the experimenter. Subjects were asked to rate pain sensations on a numerical rating scale (NRS) either during (continuous 4Hz stimulus) or immediately after electrical stimulation (single 1Hz half period pulse).

## Axon-reflex flare assessment

Skin blood flow was assessed using laser Doppler imaging (LDI, Moor Instruments LTD, Axminster, UK). The distance between the LDI scanner head and the skin was maintained at 50 cm. LDI scans were acquired sequentially over 10 minutes with each raster scan lasting 1 minute and assessing a 25 cm$^2$ skin area. Changes in skin blood flow caused by activation of unmyelinated primary sensory afferents (neurogenic mediated axon-reflex) were quantified off-line using the software package supplied by the manufacturer (MoorLDI 3.08; Axminster, UK). The area of the axon-reflex erythema (flare) was defined by the total number of contiguous pixels exceeding a threshold above baseline skin blood flow perfusion. Threshold was the mean of all pixels plus two standard deviations as determined from the baseline blood flow scan at each test site, i.e. prior to injection or iontophoresis. The area of axon-reflex flare was calculated as the number of pixels above threshold within a contiguous region around the stimulation site. All images within the LDI-sequence were analysed and the area of axon-reflex flare development depicted over time.

## Intradermal GABA injection and NKCC1 blockade

Since GABA evokes calcium responses in only a subset of DRG neuronal somata [3], the effect of intradermal GABA injection on axon-reflex skin flare and pain was examined (see S1 Fig). GABA (1mM) was injected intradermally in a volume of 100μl NaCl using a 30G insulin syringe (Becton Dickinson, Heidelberg, Germany) and the subsequent sensations and skin vasodilation were recorded for 10 minutes. Electrically-evoked pain was also examined at the site of GABA injection, before and 1 minute after injection. Intradermal injection of 100μl of 0.9% NaCl (Braun, Melsungen, Germany) served as a control.

The loop diuretic furosemide (Lasix®, Sanofi-Aventis, Frankfurt/Main, Germany) used clinically to block the inward chloride transport, was used here to manipulate intracellular chloride in cutaneous nerve terminals. Furosemide was delivered by intradermal injection (30G insulin syringe, Becton Dickinson, Heidelberg, Germany) as a 100μl bolus containing 1mg. Subject's reports of sensation and axon-reflex flare to furosemide injection were recorded for 10 minutes and compared with 0.9% NaCl injections as a control.

## NKCC1 blockade and GABA in combination on sinusoidal pain

In an attempt to disrupt the intracellular chloride concentration in sensory nerve terminals in the skin, NKCC1 blockade with furosemide and intradermal GABA were used in combination. For this protocol, 1mg furosemide (100μl Lasix®, 10mg/ml) was injected intradermally followed 25 minutes later by a 1mM GABA (100μl) injection at the same site. During the time window of 25 minutes we assumed to sufficiently prevent an NKCC1-mediated inward chloride transport via NKCC1-blockade. Also 25 minutes apart, on the contralateral arm, saline (0.9% NaCl, 100μl) and GABA (1mM, 100μl) were injected. At each site, pain ratings in response to continuous 4Hz sinusoidal current (60 s, 0.2mA) and to a 1Hz half period sinusoidal current (500 ms, 1mA) were recorded prior to any injections (baseline, t = 0), after furosemide/NaCl injection (t = 25 min), and immediately after GABA (1 min post GABA) as well as 5 minutes thereafter (5 min post GABA). For pain evaluation to sinusoidal currents at 1Hz, stimuli were presented 3 times at 1mA amplitudes with 10 second intervals between each and individual NRS as well as the average of 3 respective pain NRS ratings determined. The injection of saline or furosemide into either the left or right arm was randomised and the subject was blinded to all interventions. See S1 Fig for experimental protocol.

## NKCC1 blockade and chemically induced itch

The pruritogens histamine and codeine were used to evoke axon-reflex flare and itch sensations over a time scale of minutes and the effect of furosemide on these responses was explored. Skin sites approximately 10 cm distal to the cubital fossa on the left and right arms were injected with either 100μl furosemide (Lasix®, 10mg/ml) or 0.9% NaCl (randomized left or right). One minute after injection, histamine was applied by iontophoresis (1mA for 20 sec, i.e. 20mC) at the site of injection (S2 Fig). Skin vasodilation (LDI) and itch ratings (NRS) were recorded every minute for 10 minutes. Once sensations had subsided completely (typically about 20–30 min), the procedure was repeated on the contra-lateral forearm. The same protocol was used for assessing codeine responses on the following day. The sequence for testing histamine and codeine was randomized. The first laser Doppler scan was acquired prior to intervention (baseline) and a second scan was acquired immediately after injection of 100μl furosemide or saline. Beginning immediately after iontophoresis of either histamine or codeine an LDI sequence was acquired every minute for 10 minutes. Itch NRS ratings were recorded at 30 second intervals and subjects were blinded to all interventions.

## NKCC1 blockade and sinusoidal pain

Finally, based on the so far rather indifferent responses recorded after NKCC1 blockade, we explored whether NKCC1 blockade with furosemide affected pain ratings in a time dependent manner, in order to exclude that we may have investigated potential GABA and pruritogenic effects at the wrong time points after NKCC1 challenge. Hence, 100μl of either 1mg furosemide or 0.9% NaCl was injected intradermally into left and right forearm skin (randomized) at a site ca. 10 cm proximal to the wrist and pain ratings (NRS) to a 60 s period of continuous 4Hz sinusoidal current (0.2mA) were determined subsequently every 4 minutes for 36 minutes. As a reference, NRS ratings to a 60 s long 4Hz current (0.2mA) were determined 1–2 minutes prior to injection (baseline). In addition, NRS pain ratings to 3 bouts of 1Hz sinusoidal current (500 ms) at amplitudes of 0.2, 0.6 and 1mA (randomized) were tabulated immediately prior to injection as well as at time points 16 and 32 minutes after injection.

## Solutions

The standard solution for calcium imaging comprised: 140 mM NaCl, 5 mM KCl, 2 mM CaCl2, 1 mM MgCl2, 10 mM HEPES and pH was adjusted to 7.44 with NaOH. This imaging solution was used to dilute stock solutions to their final concentration for use on the day of each experiment. Stock solutions of Fluo8-AM (1mM), capsaicin (50mM, Sigma Aldrich), picrotoxin (10mM, Sigma Aldrich), bumetanide (100mM, Santa Cruz Biotechnology, Inc., Dallas, TX, USA) and nimodipine (200mM, Cayman Chemical, Ann Arbour, MI, USA) were dissolved in DMSO. Stock solutions of histamine dihydrochloride (200mM, Merck KGaA, Darmstadt, Germany) were made up in PBS. Stock solutions of bicuculline (100mM, from Sigma Aldrich) were made by dissolving in chloroform. $NiCl_2$ (1M, Sigma Aldrich) was held as a stock solution in distilled water. GABA (Sigma Aldrich) stock solutions of 100mM were made up fresh on the day of each experiment in DPBS.

For solutions containing nominally zero extracellular calcium, $CaCl_2$ was replaced with an equimolar amount of $MgCl_2$. A depolarising version of imaging solution containing 45mM potassium solution was prepared by replacing 40mM NaCl with an equimolar amount of KCl. Solution osmolarity was measured with a vapour pressure osmometer (Modell 5600, Wescor, South Logan, UT, USA) and adjusted with glucose to within 290–310 mOsm.

Furosemide (Lasix® 10mg/ml) is available for infusion (i.v.) in patients. Codeine phosphate hemihydrate was purchased from Ct-Arzneimittel GmbH, Berlin, Germany.

## Statistics

Statistical analyses were performed either with Statistica 7.0 (StatSoft, Tulsa, OK, USA), or GraphPad Prism (Dotmatics, Boston, MA, USA). For calcium imaging data individual ROIs (cells) were considered biological replicates (n = number of cells). Group comparisons were made after log transformation of raw fluorescence values that were subjected to repeated measurement analysis of variance (RM-ANOVA) or paired t-tests. For human axon-reflex flare data, RM-ANOVA was also used for statistical comparisons over time. Although pain and itch ratings are strictly categorical data, since the majority of values are distributed between 1 and 5, Kolmogorov-Smirnov tests indicate that NRS datasets are often normal and for these datasets parametric frequentist statistics were used. We hypothesized GABA and NKCC1 would have an impact on the endpoints "itch"/"pain" and "axon reflex flare" with statistical power of 95%. The depiction of data is indicated in the legends to figures and in most cases group data are presented as mean and standard deviation (SD). P-values are cited for all statistical comparisons.

## Figures

Figures were made with IGOR. Supplementary figures were created with PowerPoint.

## Results

### GABA evoked calcium responses in a subset of acutely isolated mouse DRG neurons

Previous reports indicate that a sub-population of DRG neuronal cell bodies respond to bath application of GABA with an increase in intracellular calcium [3, 26]. Here, we confirm this observation by demonstrating GABA-evoked calcium responses in mouse DRG neurons (Fig 1A & 1B). GABA-evoked calcium responses were observed in 336/1720 (ca. 20%) of mouse lumbar DRG neurons (diameter 21.0 ± 2.61 μm). Calcium responses to GABA had an $EC_{50}$ of 16.67μM, were mediated by an influx of extracellular calcium (Fig 1C & 1D, top) and were

blocked by nimodipine (RM-ANOVA $F_{(2,10)} = 33.5$, $p < 0.001$, $n = 6$) indicating a primary role for dihydropyridine sensitive L-type calcium channels (Fig 1C & 1D, middle). GABA evoked calcium responses were reduced in amplitude by blockade of T-type calcium channels with NiCl (1μM; paired-t-test, $t = 2.148$, $p<0.037$, $n = 47$, Fig 1C & 1D, bottom). GABA-evoked calcium responses in DRG neurons exhibited a GABA$_A$R pharmacology: they were blocked by the non-competitive, allosteric modulator picrotoxin (50μM, Fig 1E & 1F top, paired t-test, $t = 5.604$, $p<0.001$, $n = 19$) and the competitive antagonist bicuculline (50μM; Fig 1E & 1F bottom, RM-ANOVA, $F_{(2,92} = 68.49$, $p<0.001$, $n = 47$). Finally, we assessed the effect of bumetanide, i.e. inhibiting NKCC1 mediated inward chloride transport and thereby the Cl- gradient, on the GABA-induced calcium signals ($n = 122$). We saw a progressive reduction in GABA-induced calcium transients with each subsequent application of GABA (200μM) during bumetanide (25μM). Responses from the first (control) GABA application ($0.25 \pm 0.22$) were reduced to $0.03 \pm 0.05$ by the third application (RM-ANOVA, $F_{(6, 726)} = 127.1$, $p<0.001$, post hoc Bonferroni corrected t-test, $t(726) = 18.97$, $p<0.001$) and remained at this level after the fifth GABA challenge during bumetanide (Fig 1G & 1H). There was no recovery after washout of bumetanide (post hoc t-test, $t = 2.361$, $p = 0.055$). Thus, GABA-evoked calcium responses in DRG neurons exhibited a strong dependence on chloride gradients, since blockade of the NKCC1 transporter with 25μM bumetanide substantially decreased the calcium signals evoked by GABA 200μM.

To determine the phenotype of GABA-sensitive DRG neurons, pruriceptive and nociceptive stimuli in the form of histamine and the TRPV1 agonist capsaicin were tested. In an initial subset of 106 DRG neurons, a total of 39 cells responded to GABA (200μM) and a total of 39 cells responded to capsaicin (2μM). 27 of these 39 cells responded to both (25%, 27/106, Fig 1I & 1J). In a separate cohort we tested the cross-sensitivity of DRG neurons to GABA and histamine (Fig 1K). In this group of 201 DRG neurons, none of the neurons responded to both GABA (250μM) and histamine (200μM), rather 27 cells (13.4%) responded only to GABA (Fig 1L, cyan symbols) while 18 cells (9%) were responsive only to histamine (Fig 1L, magenta symbols). Since none of the GABA positive neurons responded to histamine and none of the histamine-positive neurons responded to GABA, it appears that GABA sensitive mouse DRG neurons are likely to comprise both nociceptive and non-nociceptive neurons but not histamine-sensitive pruriceptors.

Additionally, we assessed electrically induced calcium transients to 10Hz supra-threshold rectangular pulses in combination with tetrodotoxin (TTX) to dissect TTX-sensitive (TTX-s) and TTX-resistant (TTX-r) responses of DRG neurons in order to examine whether GABA-evoked calcium signals were dependent upon TTX-sensitive isoforms of voltage-gated sodium channels. Presumed nociceptive DRGs in which electrical calcium signals were not attenuated by TTX (1μM) were more likely to respond to GABA (200μM), whereas GABA-evoked calcium responses were not reduced by 1μM TTX (thus in TTX-resistant neurons) suggesting that calcium responses to GABA are not dependent upon TTX-sensitive sodium channels (paired t-test, $t = 1.510$, $p = 0.139$, $n = 40$, Fig 1M & 1N).

## Intradermal GABA does not provoke a flare response in human skin or modulate pain or flare induced by electrical stimulation

Having confirmed that GABA evoked an increase in intracellular calcium in acutely isolated mouse DRG neurons we questioned whether this finding might translate to C-nociceptor activation and neuropeptide release from sensory nerve terminals in human skin. Specifically, whether intradermal injection of GABA would cause an overt pain or itch sensation and might lead to spreading neurogenic skin flare response. We injected a 100μl volume of GABA (1mM)

intradermally on the volar surface of the forearm and compared effects with a control injection of 100μl saline (0.9% NaCl) on the opposite arm in 6 subjects (3 male, 3 female, age 46 ± 13 years). Some subjects reported brief discomfort associated with needle insertion and on occasion volume injection, however no additional sensations were reported, neither immediately after injection, nor during the 10 minute observation period thereafter. Pain reported upon GABA injection was NRS 0.8 ± 0.5 and tended to be lower than pain sensation after NaCl injection NRS 1.3 ± 0.4 (n. s.). With regard to flare responses, NaCl injection produced an axon-reflex flare response of 4.2 ± 2.8 cm$^2$ while the maximum flare area was 4.9 ± 2.6 cm$^2$ at the GABA (1mM) injection site (n.s.). There was no difference in the temporal or spatial development of the flare between GABA and NaCl injected skin sites (ANOVA interaction $F_{(5,25)}$ = 0.73, p = 0.60, Fig 2A).

We further assessed in volunteers (5 male, 5 female, age 36 ± 11 years) whether GABA / NaCl can modulate 1 minute after injection the pain magnitude induced by electrical sinusoidal stimulation. The pain ratings upon 4Hz sinusoids at 0.2mA amplitudes showed a characteristic time profile (Fig 2B) with an initial increase in NRS and a peak at 10–15 seconds followed by a gradual decline (effect of time in repeated measures ANOVA $F_{(7,63)}$ = 17.2, p<0.00001). The decline of pain rating before and after the injection of GABA / NaCl was also significant (ANOVA $F_{(1,9)}$ = 32.5, p<0.001), but did not differ between the two injections sites: peak NRS values dropped from 3.8 ± 2 to 2.1 ± 1.5 at the saline site and from 4.2 ± 1.7 to 2.1 ± 2 at the GABA injection site (ANOVA interaction, $F_{(1,9)}$ = 0.86, p = 0.38; right panel Fig 2B) most probably based on an unspecific volume effect of the injections that reduced efficacy of the electrical stimulation. There was also no significant difference in the overall time profile of NRS ratings between the two injection sites (ANOVA interaction $F_{(7,63)}$ = 1.22, p = 0.33). Thereafter we recorded the axon-reflex mediated vasodilation with laser Doppler imaging.

Pain ratings were also assessed to short bursts of C-nociceptors induced by single bouts of sinusoidal currents lasting 500 ms (half-sine shape, 1mA, 1Hz), before and 3 minutes after intradermal injections. These stimuli were delivered in triplicate at 10 s intervals and this resulted in an increase in NRS pain ratings with each repetition ($F_{(2,18)}$ = 7.5, p < 0.005, Fig 2C). Pain ratings were reduced after intradermal injections (ANOVA, $F_{(1,9)}$ = 12.8, p<0.01), but this reduction did not differ between GABA and NaCl treatment (ANOVA interaction, $F_{(1,9)}$ = 0.51, p = 0.49). Incidentally, the pain ratings to 1Hz sinusoids differed between the two injection sites with higher levels at the GABA injection site when also the pre-injection ratings were included (ANOVA $F_{(1,9)}$ = 8.38, p = 0.018) indicating a pre-existing difference independent of our intervention. Lower NRS values after both injections confirmed the effect of the volume injection, again unrelated to potential GABA-specific effects.

## Intradermal GABA alone and in combination with furosemide does not modulate electrically induced pain

In an attempt to reduce intracellular chloride in primary afferent terminals, skin sites were pre-treated with an intradermal injection of NaCl (0.9%) or furosemide (1mg) to inhibit NKCC1 chloride transport. After 25 minutes of pre-treatment, GABA (1mM) was injected to stimulate chloride efflux and pain ratings to electrical sinusoidal stimuli (single 1mA half-sine bouts and 1min continuous 4Hz stimulation at 0.2mA) tested in twelve subjects (4 male, 8 female, age 36 ± 13 years). Pain recordings were obtained at baseline, 25 minutes after NaCl / furosemide and 1 and 5 minutes after GABA injection.

### 1Hz half-sine bouts

Overall, pain ratings to 3 repetitive half-sine bouts tested 25 minutes after furosemide or NaCl injection did not differ from baseline measurements (ANOVA $F_{(1, 11)}$ = 0.2, p>0.6, Fig 2D).

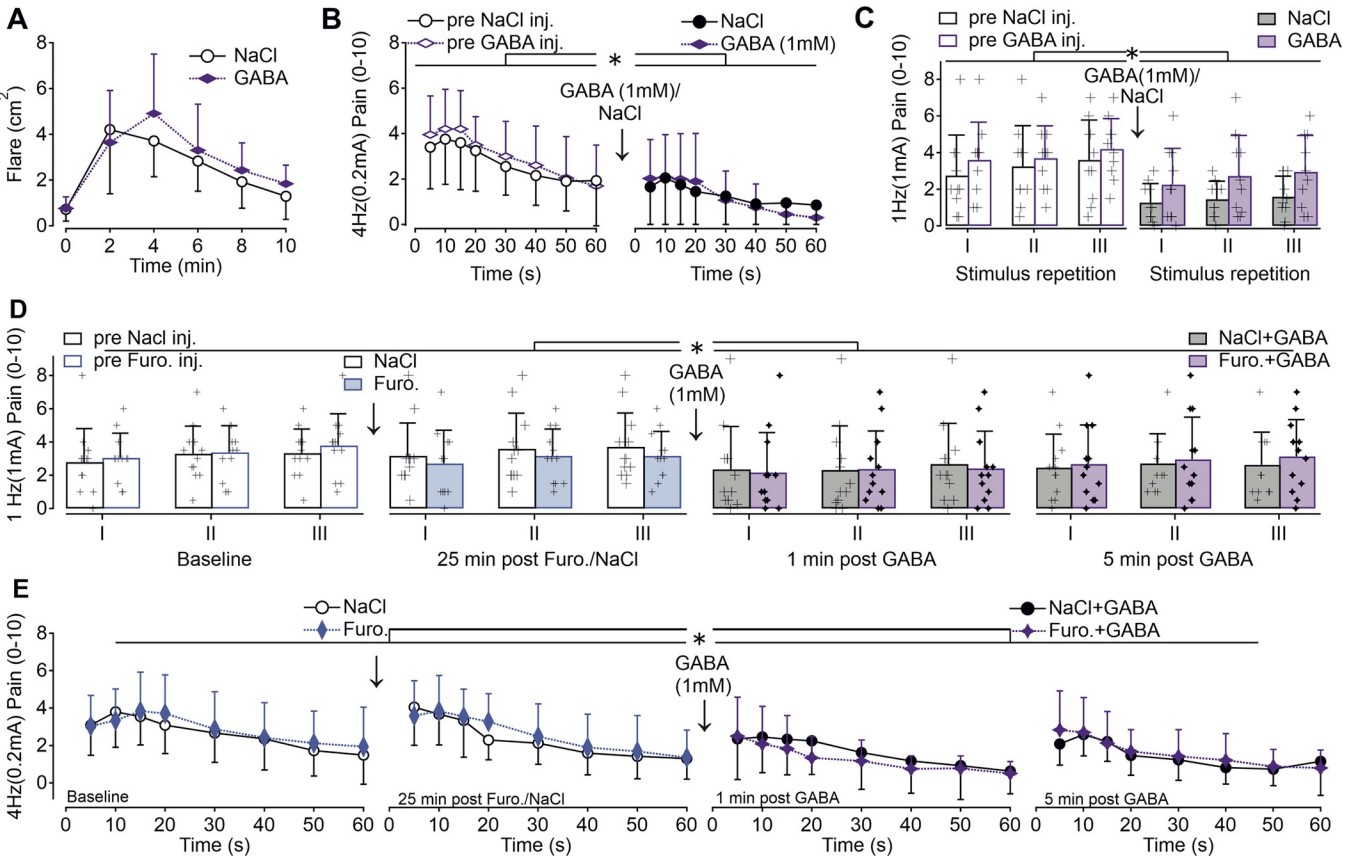

**Fig 2. The effect of intradermal injections of GABA and furosemide + GABA in combination on flare and electrical excitability.** A) Intradermal injections of 100μl GABA (1mM, purple solid symbol) caused an axon-reflex skin flare that was identical to 100 μl saline (0.9% NaCl) injections (black circle, ANOVA, n. s. p > 0.5, n = 6). B) Electrically induced pain ratings (NRS) to transdermal 0.2mA sinusoidal stimulation (4Hz) delivered to the volar forearm of n = 10 subjects and applied continuously for 60 sec before 100μl intradermal injections (left panel, open symbols) of 1mM GABA (purple rhombus) or 0.9% NaCl (black circle) and 1 minute after intradermal injections (right panel, solid symbols) of GABA (purple) and NaCl (black). Prior to injection a maximum pain of approx. NRS 4 developed within 15 sec of sinusoidal stimulation and continuously declined throughout the 60 sec recording time (left panel). Following 100μl injections of GABA or NaCl, electrically evoked pain was significantly lower in comparison to baseline condition prior injections (ANOVA, asterisk p < 0.001) with an average of NRS 2 during 15 sec at both treated skin sites (ANOVA, n.s. compared to NaCl). C) Bar chart representing the average pain ratings to a 1Hz (1mA) half period sinewave pulse delivered to volar forearm skin (n = 10). Stimuli were applied repetitively 3 times (I-III) at 10 sec intervals before injection (naïve skin) of 100μl GABA (1mM) or NaCl (open columns, left) and 3 min after injection (filled columns, right). Estimated pain magnitude to 1Hz stimuli increased significantly across the 3 stimulus repetitions (ANOVA, p < 0.005) and was significantly higher in naïve skin compared to injected skin sites (ANOVA, p < 0.01, marked by asterisk). Individual recordings are indicated by +. D) Pain ratings to 1Hz half period sinewave pulses of 1mA amplitude were recorded at baseline (before injection, open columns) and 25 minutes after 100μl injections of 1mg furosemide (blue columns) or NaCl (white columns, n = 12). GABA (100μl, 1mM) was injected into furosemide-/NaCl-treated skin and pain NRS to the 1Hz stimulus monitored immediately after injection (dark grey/purple columns) and to a further bout of electrical stimulation 5 minutes later. Pain ratings were not significantly different between furosemide 1mg and NaCl (0.9%) treatment and compared to baseline (ANOVA, p > 0.6). Injection of 1mM GABA caused significantly less electrically evoked pain in both furosemide and NaCl skin (ANOVA, p < 0.05, marked by asterisks) and thus was not significantly different between the sites (n.s.). Similarly, pain ratings recorded 5 min after GABA injections was identical at the furosemide and saline-sites (n.s.). Individual recordings of each subject are indicated by +. E) Assessment of pain ratings to sinusoidal 4Hz stimulation delivered with 0.2mA amplitudes continuously for 60 seconds and at the same time points as in (D), n = 12 subjects. Injection of furosemide 1mg (blue diamond) had no significant effect on pain magnitude compared to NaCl (black open circles) and were virtually identical to baseline condition (prior to injections). Administration of 100μl GABA 1mM to both furosemide and NaCl treated sites (solid black and purple symbols) diminished sinusoidal 4Hz pain significantly (ANOVA, p < 0.02, marked by asterisk), but reduction of pain ratings to GABA 1mM was not significantly different between furosemide and NaCl treatment (right panels, n.s.).

The subsequent intradermal injection of GABA reduced pain ratings (ANOVA $F_{(3, 33)} = 3.07$, p = 0.041), but this effect was independent of preceding NKKC1 blockade by 1mg furosemide (ANOVA interaction $F_{(3, 33)} = 1.43$, p = 0.25).

## 4Hz 1 minute sinusoids

Similar to the effect on NRS ratings to 1Hz sinusoids, a generalised reduction in NRS values was observed following intradermal GABA (1mM) injection (ANOVA F(1, 11) = 13.76, p = 0.003, Fig 2E) irrespective of prior injection with furosemide or NaCl (ANOVA interaction F(1, 11) = 0.56, p = 0.47). Pain ratings following injection of furosemide and NaCl did not change (ANOVA F(1, 11) = 0.45, p>0.5). Immediately after GABA injection (1 min), 4Hz sinusoidal 0.2mA pain was not different between furosemide and NaCl (ANOVA, F(1,11) = 1.28, p>0.2) and the second test after GABA at 5 min resulted in no additional pain alteration either (ANOVA F(1,11) = 0.98, p>0.3). As previously observed, only a volume effect on electrically-evoked pain was observed, whereas no substance-specific effects following intradermal injection of furosemide or GABA, individually nor in combination, were found.

## Itch ratings to histamine and codeine iontophoresis are not affected by NKCC1 blockade

In vitro data suggests that $GABA_A R$ activation stabilises the excitability of C-fiber axons during ongoing low frequency electrical stimulation via NKCC1-mediated inward chloride transport [7]. To test whether this in vitro finding might translate to alterations in human sensation in vivo we used chemical itch as a model of prolonged C-fiber activity over minutes [27]. Histamine (19 subjects, 9 male, 10 female, age 35 ± 11 years) and codeine (12 subjects, 4 male, 8 female, age 36 ± 13 years) were applied individually (14 hours apart) to the forearm skin by iontophoresis and both produced long-lasting sensations of itch over several minutes (Fig 3A). Chemically induced itch ratings peaked approximately 1.5 min after iontophoresis at NRS 2.1 ± 1 for histamine (Fig 3A, left panel) and at NRS 2.7 ± 1.5 for codeine (Fig 3A, right panel) and subsequently declined to NRS 0.5 ± 0.6 for histamine (ANOVA F(19,342) = 15.36, p<0.001) and NRS 0.3 ± 0.4 for codeine (ANOVA F(19,190) = 17.61, p<0.001). Blocking NKCC1 with a preceding injection of furosemide (1mg) to disrupt chloride regulation in sensory afferents had no effect on the magnitude nor on the time-course of NRS itch ratings to histamine (left panel, Fig 3A; ANOVA (F(1,18) = 0.71, p = 0.41, interaction over time F(19,342) = 0.83, p = 0.67) nor to codeine (right panel, Fig 3A; ANOVA NRS F(1,10) = 0.93, p = 0.36, interaction over time F(19,190) = 0.54, p = 0.94).

## Flare responses to histamine and codeine are larger after NKCC1 blockade

Before examining interactions between chemical pruritogens and furosemide on C-nociceptor mediated flare responses, the effect of intradermal furosemide (0.1 and 1mg) was examined in isolation. In 8 subjects (4 male, 4 female, age 47 ± 10 years) furosemide injection did not induce a visible wheal response nor any sensation of itch. However, using laser Doppler skin blood flow imaging, an increase in vasodilation was observed that was larger than that after intradermal NaCl (Fig 3B, left panel) for 0.1mg (4.5 ± 2.7cm$^2$ versus NaCl 3 ± 1.9 cm$^2$, ANOVA F(1,7) = 7.98, p = 0.026) and 1mg furosemide (6.5 ± 3cm$^2$ versus NaCl 2.7 ± 1.9 cm$^2$, ANOVA F(1,7) = 21.39, p = 0.002).

The effect of histamine and codeine iontophoresis on flare were then examined at the furosemide and NaCl injected sites. For histamine, skin flare areas at the furosemide site were larger than at the saline injected site (15 ± 4.5 cm$^2$ versus NaCl 12 ± 4.3 cm$^2$, ANOVA (F(1, 18) = 5.25, p = 0.034) and the time course was prolonged (ANOVA interaction over time F(10,180) = 3.78, p<0.0001; centre panel, Fig 3C). Similarly, skin flare responses to codeine were marginally larger at furosemide compared to NaCl treatment (13 ± 4 cm$^2$ versus NaCl 10 ± 3cm$^2$, ANOVA F(1, 11) = 4.87, p = 0.0496) and the time course was also prolonged

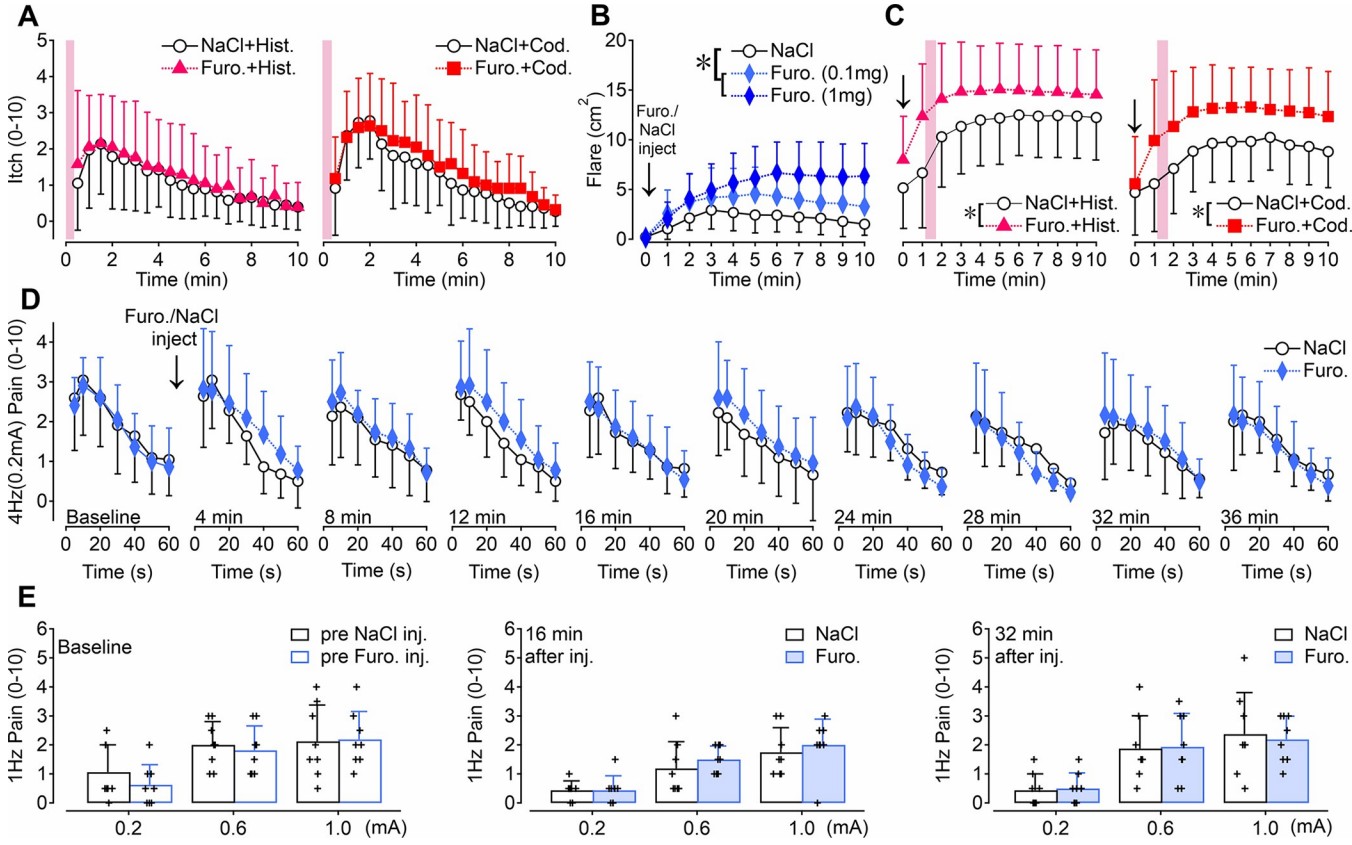

**Fig 3. The effect of NKCC1 blockade by intradermal furosemide on itch, pain and axon reflex flare.** A) Assessment of itch (numeric rating scale, NRS 0–10) in response to iontophoresis (20mC) of histamine-dihydrochloride (left panel, triangles, n = 19) or codeine-phosphate (right panel, squares, n = 12) delivered 1 minute after intradermal injections of 100μl furosemide (1mg, filled symbols) or saline (0.9% NaCl, open circles) into the left and right volar forearm, respectively. Both pruritogens evoked maximum itch about 2 min after iontophoresis that declined significantly within the 10 min recording period (ANOVA, p < 0.001). No significant difference of itch magnitude was recorded between the pruritogens and the furosemide / NaCl pre-treatment respectively (n.s.). B) Axon-reflex flare (cm$^2$) assessed after intradermal injection (indicated by arrow) of 100μl furosemide 0.1mg and 1mg (blue diamonds, n = 8, left panel) and 0.9% NaCl (black open circles). Furosemide evoked a dose-dependent and significantly larger flare area in comparison to NaCl (ANOVA, p > 0.02 (0.1mg) and p < 0.005 (1mg), asterisk, left panel). C) Injection of 1mg furosemide (indicated by arrows) evoked a significantly larger axon-reflex flare response to 20mC iontophoresis (indicated by pink columns) of histamine (pink triangles, centre panel) or codeine (red squares, right panel) when compared to NaCl pre-treatment (black open circles) throughout the 10 min recording period (ANOVA, p < 0.05, marked by asterisks). D) Pain ratings (NRS) to sinusoidal current at 4Hz at an amplitude of 0.2mA delivered continuously for 1 minute before (baseline) and after intradermal injection of 1mg furosemide (blue diamonds) or 0.9% NaCl (black open circles), recorded every 4 minutes for 36 min (n = 9). No significant difference of electrically-induced pain was recorded between furosemide and saline treatment throughout the observation period (n.s.). Note that pain ratings during 30–60 sec of stimulation tended to be lower at the furosemide compared to the saline sites at minute 24 and 28 after injection. E) Pain ratings (NRS) to 1Hz sinusoidal stimulation delivered at current intensities of 0.2, 0.6 and 1mA to the forearm skin sites before (pre-)injection (baseline, open columns, n = 8) and after injection of 1mg furosemide (light blue) or 0.9% NaCl (black outline columns) at 16 min (middle panel) and 32 min (right panel). Pain increased significantly with enhanced currents (ANOVA, p < 0.005) but no significant difference was recorded after NKCC1 blockade (1mg furosemide) at 16 and 32 minutes (n.s.).

(ANOVA interaction over time $F_{(10,110)}$ = 2.65, p<0.01; right panel, Fig 3C). Taken together, histamine and codeine induced skin flare responses were enhanced after local NKCC1 blockade with furosemide, but the level of the accompanying itch ratings were not altered by preceding furosemide injection.

## Pain ratings to electrical sinusoids were not affected by NKCC1 blockade with furosemide

The effect of intradermal furosemide (1mg) on electrical 4Hz sinusoidal pain was determined in 9 subjects (5 male, 4 female, age 44 ± 11 years) in 4 minute intervals over a period of 36

minutes. Before furosemide injection, pain ratings to 1 minute 4Hz sinusoidal 0.2mA current amplitude stimulation peaked at around 10 sec with an NRS of 3 ± 1 and declined subsequently to NRS 1.1 ± 1 at the end of the 60 s stimulation period ('Baseline', Fig 3D). Intradermal furosemide or NaCl injection (downward arrow in Fig 3D) did not change pain ratings to the 4Hz stimulation as between NaCl and furosemide sites neither the average ratings differed throughout the 36 minutes (ANOVA interaction $F_{(9, 72)}$ = 1.19, p = 0.32) nor the time course of pain during the 60 s stimulus (ANOVA interaction $F_{(54, 432)}$ = 1.18, p = 0.195). This result suggests that the ability of nociceptors to generate action potentials at a low rate of 4Hz was unaffected by NKCC1 blockade.

Finally, pain ratings to electrical half period 1Hz sinusoidal cycles (500ms pulse) of 0.2, 0.6, 1mA were recorded from NaCl and furosemide sites 16 and 32 minutes after treatment. Prior to furosemide or saline injection (baseline), pain ratings to single bouts of 500ms sinusoids increased intensity-dependently (ANOVA $F_{(2, 14)}$ = 7.66, p = 0.006; leftmost panel Fig 3E). Injection of furosemide or NaCl had no effect on overall pain ratings at 16 minutes (centre panel, Fig 3E; ANOVA $F_{(1, 7)}$ = 0.86, p = 0.39) and 32 minutes (rightmost panel Fig 3E, ANOVA $F_{(1, 7)}$ = 0.003, p = 0.95) or the current intensity-dependent increase of pain (Fig 3E; ANOVA interaction $F_{(2, 14)}$ = 0.57, p = 0.58). This suggests that the ability of nociceptors to encode different intensities of 500ms depolarizing stimuli was not affected by NKCC1 blockade.

## Discussion

We studied to which extent depolarizing responses by GABA reflected as calcium transients in sensory neurons in vitro might translate into either a flare response or modulation of pain or itch when applied intradermally in people. The impetus here was to assess the effect of GABA on C-fiber mediated sensations of pain by leveraging recently developed sinusoidal electrical stimulus paradigms [28]. Despite confirming previous reports of GABA-evoked calcium signals in isolated DRG neurons, GABA was largely without effect when injected into human skin. The intracellular chloride concentration determines the direction of GABA mediated ion fluxes. Reducing the chloride gradient by blocking the NKCC1-mediated inward chloride transport using intradermal furosemide was per se without effect on human ratings to chemical itch or electrical pain and also did not modulate GABA responses in our experimental human models indicating the complexity of translating results from well-controlled cellular approaches to human in-vivo conditions.

The observation here that GABA evokes calcium responses in isolated DRG neurons via $GABA_AR$ has been previously well established [3, 8, 26, 29]. We extend this finding by demonstrating that GABA-evoked calcium transients persist during blockade of TTX-sensitive voltage-gated sodium channels (Fig 1M & 1N). This is consistent with whole cell [29] and perforated patch recordings showing that depolarizing $GABA_AR$ currents in DRG neurons are insufficient to evoke action potentials [8, 9, 16, 30]. The inability of GABA to generate action potentials in DRG neurons is also consistent with the lack of sensations accompanying intradermal GABA observed here and reported previously [29, 31]. Thus, the lack of action potential generation accompanied with an absent neuropeptide release from terminal endings of sensory afferent neurons is in line with the lack of the widespread neurogenic axon-reflex flare in humans.

In addition to a lack of overt sensations associated with intradermal GABA, there was no evidence for the modulation of chemically-evoked itch or electrically-evoked pain by GABA (Fig 3). This is contrasting older results showing that iontophoresis of the loop diuretics frusemide and bumetanide into human skin can reduce flare and itch upon subsequent injections

of histamine [32]. Independent of potential GABA effects, calcium-activated chloride channels, in particular anoctamin 1 have been shown to contribute to chloroquine-induced action potential generation in pruriceptors and scratching behaviour in mice [33], suggesting that reduction of the chloride gradient by loop diuretics should impair such an activation. However, in contrast to chloroquine, histamine-induced scratching was unchanged in anoctamin 1-deficient mice [33] leaving the role of chloride currents for the chemical activation of pruriceptors ambiguous. Our failure to observe inhibitory effects of NKCC1 blockers on itch could also be linked to a different application methodology of histamine: the instantaneous increase of histamine concentration upon intradermal injection [32] might have been more suited to reveal inhibitory effects of a reduced chloride gradient as compared to the rather slow increase by our iontophoresis technique. In addition, in pilot experiments we observed a very distinct sensation of itch for the potent NKCC1 inhibitor torasemide (therefore omitted this compound from the protocol). Even after the injection of furosemide we found a small increase in axon reflex erythema (see Fig 3). Thus, we cannot exclude secondary effects in-vivo that may be unrelated to NKCC1 inhibition.

*In vitro*, 200μM GABA applied to DRG neurons predominantly reduced the volley of action potentials produced during ramp depolarization [9] and reduced slope and amplitude of the action potentials, which is in-line with its shunting effect and depolarization-induced inactivation of voltage-sensitive sodium channels. On the other hand, few DRG neurons also increase their firing activity after this GABA treatment [9], potentially supporting differential effects among nociceptor classes. In particular excitation of mechano-insensitive "silent" nociceptors, that have a TTX-resistant peripheral conduction [34] may be facilitated when membrane potential is shifted close to the activation threshold of TTX-resistant voltage-sensitive sodium channel NaV1.8. However, we did not detect any modulation of chemically evoked itch or of electrically-induced pain. Cutaneous axons responding to histamine are peptidergic "silent" nociceptors in humans [27, 35, 36] and as demonstrated in Fig 1L, calcium responses to GABA and histamine were observed in mutually exclusive populations of mouse DRG neurons. Moreover, electrical sinusoids (4Hz) activate superficial C-fiber terminals [37] and the threshold for sensation using these electrical sinusoids matches that for a flare response [28], which might support the idea that our chemical and electrical tests in human skin did not include the relevant GABA$_A$R-positive nociceptor class. However, 4Hz electrical sinusoids also readily activate mechano-sensitive polymodal nociceptors in humans [28] and importantly, the single bouts of depolarizing electrical stimuli (500ms, 1Hz) particularly activate polymodal nociceptors while sparing mechano-insensitive chemo-nociceptors [38]. Thus, our electrical stimulation paradigms activate all classes of epidermal nociceptors in humans and thus our negative results of GABA on electrically- and chemically-evoked responses cannot be explained by targeting different nociceptor classes. Moreover, the translation of nociceptors classes between rodents and human is currently under debate and the segregation of peptidergic and non-peptidergic nociceptors may not be useful in humans [39].

The key limitation of our study is the lack of direct electrophysiologic recordings of neuronal GABA effects. However, we were mainly interested in the effects on sensory endings and axonal recordings are technically challenging and limited to in-vitro conditions [40] When compared to our *in vitro* results in mice, we cannot exclude that differences in local expression of the targeted receptors, ion-channels or co-transporters between the soma of the DRG neurons and their peripheral sensory ending also might have contributed. Moreover, the GABA current equilibrium potential has been found about 20 mV more hyperpolarized in human as compared to rat DRG neurons further limiting direct translation.

Our in-vivo approach in humans allowed for psychophysics and axon reflex flare assessment, however, the intradermal application of blockers and mediators was invasive and may

have changed the geometry of the sensory endings by the volume injected into the skin. Indeed, we observed acutely reduced pain ratings to electrical stimulation (Fig 2B & 2C) in equal measure for both NaCl vehicle and GABA. most likely as a result of a reduction in the effective electric field strength for tissue beneath the low resistance fluid bleb or a possible compromise in the structural integrity of nerve terminals. Moreover, the limited knowledge about pharmacokinetics of the intradermally injected GABA including degradation and uptake, but also desensitization of GABA-A receptors on the sensory endings is mainly unclear. Finally, the opposing effects of GABA on nociceptor excitability may have at least partly cancelled out one another.

While successful in-vitro and in ex-vivo nerve excitability recordings, our efforts to disrupt chloride homeostasis in sensory nerve endings in human skin using intradermal injection of GABA in combination with the NKCC1 blocker furosemide failed to elicit an overt sensation, did not affect the magnitude of evoked pain or itch, and had no influence on basal vasodilation or axon-reflex mediated increase of skin blood flow. This indicates the complexity of translating cellular mechanisms into human in-vivo approaches. Moreover, our results in healthy volunteers do not exclude a major role of NKCC1 in chronic neuropathic conditions [41] or following axotomy [42] or under inflammatory conditions that increase NKCC1 activity [17]. It is therefore interesting that patients with chronic kidney failure that receive a furosemide therapy report lower itch ratings [21]. Hence, the negative result in our translational approach to manipulate experimentally evoked C-nociceptor excitation based on healthy volunteers do not exclude a clinical role of NKCC1 in chronic pain or itch.

## Supporting information

**S1 Fig. Experimental protocol as a flowchart to assess the effect of NKCC1 blockade on GABA-induced responses.** A) GABA or NaCl was injected into each respective forearm and axon-reflex flare (skin vasodilation) recorded using LDI for 10 minutes. Corresponding to Fig 2A. B) Electrically-evoked pain responses to 1Hz half period sinusoidal current and 4Hz sinusoidal current were assessed before and 1 minute after injection in verum and control treated forearms. Corresponding to Fig 2B. C) Furosemide (1mg) or NaCl were injected intradermally in either arm. Corresponding to Fig 2C. D) At baseline (prior to any injections) and 25 minutes after furosemide / NaCl injection, pain ratings in response to continuous 4Hz sinusoidal current and 1Hz half period sinusoidal current were assessed. Corresponding to Fig 2D. E) GABA (1mM) was injected at the same site of previous furosemide and NaCl injection (randomized forearms) and pain responses to 1Hz half period sinusoidal (1mA) and 4Hz sinusoidal (0.2mA) currents were again assessed at 1 minute and 5 minutes after GABA injection. Corresponding to Fig 2E.
(TIF)

**S2 Fig. Experimental protocol for the assessment of NKCC1 blockade and chemically-induced itch.** A) Skin sites approximately 10 cm distal to the cubital fossa on the left and right arms were injected with either 100μl furosemide or NaCl (randomized left or right). One minute after injection, histamine was applied by iontophoresis at the site of injection. Itch ratings (NRS) were recorded. Corresponding to Fig 3A. B) Skin sites approximately 10 cm distal to the cubital fossa on the left and right arms were injected with either 100μl furosemide at two different concentrations (0.1mg, light blue or 1mg, dark blue) or NaCl (0.9%). Skin vasodilation (LDI) was recorded. Corresponding to Fig 3B. C) One minute after injection of Furosemide or NaCl, histamine or codeine was applied by iontophoresis at the site of injection. Skin vasodilation (LDI) was recorded every minute for 10 minutes. 24 hours later, the same protocol was used for assessing the other (histamine or codeine) substance. D) Pain ratings (NRS)

to sinusoidal current at 4Hz at an amplitude of 0.2mA delivered continuously for 1 minute before (baseline) and after intradermal injection of 1mg furosemide (blue diamonds) or 0.9% NaCl (black open circles), recorded every 4 minutes for 36 min (n = 9). E) Pain ratings (NRS) to 1Hz sinusoidal stimulation delivered at current intensities of 0.2, 0.6 and 1mA to the fore-arm skin sites before injection and 16 and 32 minutes after injection of 1mg Furosemide or NaCl.
(TIF)

## Acknowledgments

The authors are grateful to Anja Bistron for technical support and indebted to each and every volunteer for their patient and willing cooperation.

## Author Contributions

**Conceptualization:** Roman Rukwied, Richard Carr.

**Data curation:** Kyra Sohns, Anna Kostenko, Marc Behrendt, Roman Rukwied.

**Formal analysis:** Kyra Sohns, Roman Rukwied.

**Funding acquisition:** Martin Schmelz, Roman Rukwied, Richard Carr.

**Investigation:** Kyra Sohns, Anna Kostenko, Marc Behrendt, Roman Rukwied, Richard Carr.

**Methodology:** Kyra Sohns.

**Project administration:** Kyra Sohns, Roman Rukwied, Richard Carr.

**Supervision:** Martin Schmelz, Roman Rukwied, Richard Carr.

**Writing – original draft:** Kyra Sohns, Martin Schmelz, Richard Carr.

**Writing – review & editing:** Kyra Sohns, Anna Kostenko, Martin Schmelz, Roman Rukwied.

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
