## [Decision Letter · Decision Letter 0]

2 May 2024

PONE-D-24-10034Depolarization of mouse DRG neurons by GABA does not translate into acute pain or hyperalgesia in healthy human volunteersPLOS ONE

Dear Dr. Schmelz,

Thank you for submitting your manuscript to PLOS ONE. After careful consideration, we feel that it has merit but does not fully meet PLOS ONE’s publication criteria as it currently stands. Therefore, we invite you to submit a revised version of the manuscript that addresses the points raised during the review process. It will be essential for you to respond to all of the reviewers' comments, but it is particularly important that you can successfully address: reviewer one's concerns about clarity/statistics and reviewer two's concern that the lack of effect of GABA could be due to the possibility that the chloride reversal potential is not above threshold for an AP (a significant consideration of this idea must be developed in the discussion).

We look forward to receiving your revised manuscript.

Kind regards,

Peter Wenner

Academic Editor

PLOS ONE

Journal Requirements:

"The study was supported by the German Research Foundation (Deutsche Forschungsgemeinschaft, https://www.dfg.de/en/research-funding/funding-opportunities), grant project 397846571 (RR) and FOR 2690 (MS, RR) and SFB 1158 (RC, MS)."

3. In the online submission form, you indicated that [The data underlying the results presented in the study are available from the authors upon request.]. 

5. We note that S1 and S2 Figures in your submission contain copyrighted images. All PLOS content is published under the Creative Commons Attribution License (CC BY 4.0), which means that the manuscript, images, and Supporting Information files will be freely available online, and any third party is permitted to access, download, copy, distribute, and use these materials in any way, even commercially, with proper attribution. For more information, see our copyright guidelines: http://journals.plos.org/plosone/s/licenses-and-copyright.

a. You may seek permission from the original copyright holder of S1 and S2 Figures to publish the content specifically under the CC BY 4.0 license. 

Reviewers' comments:

Reviewer's Responses to Questions

**Comments to the Author**

1. Is the manuscript technically sound, and do the data support the conclusions?

Reviewer #1: Yes

Reviewer #2: Partly

2. Has the statistical analysis been performed appropriately and rigorously? 

Reviewer #1: I Don't Know

Reviewer #2: Yes

3. Have the authors made all data underlying the findings in their manuscript fully available?

Reviewer #1: No

Reviewer #2: Yes

4. Is the manuscript presented in an intelligible fashion and written in standard English?

Reviewer #1: Yes

Reviewer #2: Yes

5. Review Comments to the Author

Reviewer #1: The paper by Sohns et al. investigated the effects of GABA in the periphery, on rodent neurons and on humans. The work on rodent neurons slightly extends existing knowledge, the human experiments represent the novelty of the paper.

Major:

- The experiments were not preregistered as a trial, and therefore the protocol and number of subjects not laid down in a public database before the investigation. Nevertheless, the experiments are approved by an ethics committee, which requires to provide hypotheses, endpoints, and a power calculations. None of this is provided in the manuscript. This should be explicitly mentioned for every question, and clearly distinguish preset and adequately powered hypotheses and those which are exploratory and might have a less favorable type I and II error.

This includes the experimental timeline, which is not easy to understand. E.g. who of the subjects in Fig. S2 proceeded from B to C, and who from B to D. A flowchart might help making this more accessible

- “The inability of GABA to generate action potentials in DRG neurons is also consistent with the lack of sensations accompanying intradermal GABA observed here and reported previously (27)”. Although the pain directly induced by GABA might be minimal, and this might explain why modulation of evoked please was investigated: Please provide data on the pain caused by GABA vs. control injection, in particularly given the failure of the only published human study cited (citation 27) to do so.

Minor:

- Figure 3a: In case the SD is negative, the data are probably skewed. The authors need to decide whether non-parametric visualization is more suitable.

- Line 187: ‘to less than 10% of the initial fluorescence ‘should rather be to less than 10% increase above the initial …

- Independent repetitions of cellular experiments should be mentioned, in addition to the cell count. E.g. in line 358, assuming 100 cells in a field of view and a 20% response rate to GABA, 19 cells might well be a single run of the experiment, which I would consider highly questionable.

- Rundown of GABA responses with butemetanide cannot be judged without an identical protocol without this substance.

- Line 277: Why where experiments single and not double blinded?

- Many values are provided with a precision which is probably not justified by the measurement, and a digit less might be the better

- Fig 1B. Is the concentration response constructed only from protocols like in Fig 1A, which has, despite tachyphylaxis in other panels, starts with a high concentration

- Fig 1I. Legend mentions 145 mM KCl, which is probably 45 mM as mentioned in the methods

- Fig 1J) A Venn diagram seems more suited to visualize the populations than a bar chart

- Fig 2 C,D,E: Is a scale up to 12 reasonable for a measure in the range 0-10?

- Fig 3B: Why did half of the skin blood flow increase in the furosemide pretreated spots occur before the iontophoresis?

Reviewer #2: This manuscript addresses the question of whether modulation of cutaneous terminals of primary somatosensory neurons by GABA alters human responses to stimuli that evoke mild to moderate pain or itch. This is a question of potential clinical importance as peripheral GABA signaling might provide opportunities to treat pain and itch caused by peripheral insults. Using dissociated mouse DRG neurons and Ca-imaging, the authors replicate previous studies indicating that GABA depolarizes DRG neuron somata sufficiently to increase intracellular Ca2+ that depends upon a high concentration of intracellular Cl- maintained by NKCC1 pump activity, and that GABA sensitivity correlates with TTX insensitivity (the sensory neurons more likely to be nociceptors or pruriceptors). They then describe various experiments with human volunteers to determine whether excitatory responses to GABA injected into the skin alter neurogenic inflammation, pain, or itch (apparently for the first time, as no references to previous studies involving cutaneous injection of GABA are cited). The most important finding is that GABA injection failed to produce pain or itch, and that injection of neither GABA nor furosemide (which inhibits NKCC1 to reduce intracellular Cl- concentration and hyperpolarize the Cl- equilibrium potential, ECl) altered the volunteers’ ratings of pain or itch induced by other cutaneous stimuli. While these are noteworthy findings and the methods are solid, I have concerns about how the experimental design and results are interpreted, which should be considered when revising the manuscript.

1. The manuscript stresses the “contrast” and “surprising” difference between the observations of GABA-evoked Ca2+ transients in dissociated mouse DRG neurons and the lack of GABA effects upon injection into human skin. Ignored completely are well-established principles about how a ligand such as GABA can both depolarize a neuron if membrane potential is below the reversal potential for Cl- and produce a net inhibitory effect via a decrease in input resistance (Rin) and shunting of excitatory currents. In nociceptive and pruriceptive DRG neurons, ECl is often depolarized relative to resting membrane potential (RMP) but hyperpolarized relative to AP threshold. For example, if a dissociated soma or peripheral process has an RMP of -65 mV, ECl of -40 mV, and an AP threshold of -35 mV (all of which have been reported for DRG neuron somata), opening of Cl- channels by themselves will not be able to depolarize RMP to the AP threshold. In addition, the increase in Cl- conductance will reduce Rin, decreasing the amplitude of sensory generator potentials and spontaneous depolarizing fluctuations of RMP, and the Cl-mediated depolarization that approaches (but cannot exceed) ECl will inactivate voltage-gated Na+ channels. Thus, there is no reason for surprise at the lack of human GABA-evoked pain and itch when extrapolating from the observations of GABA-evoked Ca2+ transients in dissociated DRG neuron somata.

2. The fact that the Ca2+ transients are relatively small suggests that the GABA-evoked depolarizations are not large enough to activate a substantial fraction of voltage-gated Ca channels and thus that these depolarizations usually fail to exceed AP threshold. A major limitation of the manuscript is that no electrophysiological measurements were made, so none of these potentials (RMP, ECl, AP threshold, GABA-evoked depolarization relative to AP threshold) are actually known. Of course, these values are not yet obtainable from the cutaneous terminals of rodent or human DRG neurons, but the inhibitory principles are clear and should be discussed as part of the experimental design and interpretation of the results.

3. Similarly, the results do not really suggest that “pharmacological intervention of GABAAR signaling in human skin is not … promising.” This is because the study is mainly designed to test interventions that target depolarizing effects of GABA. The work of Gamper, Du and colleagues has provided strong evidence that, at least in the DRG, even when depolarizing a nociceptor, GABA has strong inhibitory effects. In principle, the same mechanisms could operate on conduction through peripheral branch points and on the effectiveness of generator potentials in terminals.

4. This study did not examine conditions that might optimize inhibitory effects of GABA in cutaneous receptive fields. The closest it came to testing this idea was in the experiments of Fig.2D,E, examining the effects of furosemide pretreatment and GABA injection on pain evoked by electrical stimulation. There was a lack of apparent analgesic effects from GABA, but this might be because the electrical test stimuli produced only modest pain that was further attenuated by the injection volume effect seen in all the human experiments in this manuscript. Furthermore, the effectiveness of peripheral injections of GABA may be limited by desensitization of GABA receptors, and by potentially rapid uptake and degradation of injected GABA within the skin.

5. Because no pro-algesic effects of GABA were found in this study (Figures 2, 3), the furosemide experiments (expected to reduce the depolarizing effects of GABA) provide very limited information. However, furosemide should enhance the inhibitory effects of GABA, so the cited study reporting itch reduction by furosemide (line736) supports the possibility that enhancement of GABA effects might be therapeutically useful. Again, it would have been useful also to explicitly test whether peripheral GABA injection can have inhibitory effects on pain or itch, and perhaps to test the effects of blocking transport of Cl- out of the cells (in addition to their use of furosemide to block transport of Cl- into the cells).

MINOR COMMENTS

Line 53 – Furosemide should reduce the outwardly directed Cl- gradient, but brief applications of GABA should mainly reveal the effects of this gradient reduction, not by itself cause “depletion” of the gradient.

Line 141 – poly-L-lysine

Line 191 – what is the duration of each GABA application?

Line 352 – why show a weak effect of Ni2+ (n=47, p=0.037) rather than the strong effect mentioned from the highly specific T-type VGCC blocker TTA-P2?

Line 360 – the important effect is inhibiting transport of Cl- out of the cell to change the Cl- gradient, not the small current mediated by NKCC1 transporters.

Lines 414-418 – the responses should be higher in the neurons with a lower, not higher, TTX sensitivity, if I understand this experiment correctly. Also, which traces are being referred to?

Line 436 – electrically evoked calcium signals (not “electrical calcium signals”)

Line 704 – a reference is needed for this statement about mast cell degranulation

Line 729 – the observation that human DRG neurons have a more hyperpolarized ECl than rodent DRG neurons should be mentioned (Zhang et al., PMID: 26415765).

6. PLOS authors have the option to publish the peer review history of their article (what does this mean?). If published, this will include your full peer review and any attached files.

Reviewer #1: No

Reviewer #2: **Yes: **Edgar T. Walters

---

## [Author Response · Author response to Decision Letter 0]

31 May 2024

To whom it may concern, 

We thank you very much for the constructive comments and hope to have amended all aspects adequately. We would especially like to point out that we feel this revision, as per your suggestions, has improved the manuscript and are grateful for the opportunity to review the oversights made in the first submission. We hope that our responses in the Response to Reviewers document in which we have addressed each comment (also submitted) find you well.

All the best, 

Kyra Sohns

---

## [Decision Letter · Decision Letter 1]

2 Jul 2024

PONE-D-24-10034R1Depolarization of mouse DRG neurons by GABA does not translate into acute pain or hyperalgesia in healthy human volunteersPLOS ONE

Dear Dr. Schmelz,

Thank you for submitting your manuscript to PLOS ONE. After careful consideration, we feel that it has merit but does not fully meet PLOS ONE’s publication criteria as it currently stands. Therefore, we invite you to submit a revised version of the manuscript that addresses the points raised during the review process.

The authors have responded well to the reviewers' comments and the manuscript is much improved. However, the authors will need to make minor adjustments in the text of the results that refer to Figure 3C-E (3E is not referred to, and 3D appears to be referred to as 3C, there is no downward arrow that was referred to in the text). In addition there are minor errors in spelling in the body of the text. Please adjust these minor changed and resubmit the manuscript.

We look forward to receiving your revised manuscript.

Kind regards,

Peter Wenner

Academic Editor

PLOS ONE

Journal Requirements:

Reviewers' comments:

Reviewer's Responses to Questions

**Comments to the Author**

1. If the authors have adequately addressed your comments raised in a previous round of review and you feel that this manuscript is now acceptable for publication, you may indicate that here to bypass the “Comments to the Author” section, enter your conflict of interest statement in the “Confidential to Editor” section, and submit your "Accept" recommendation.

Reviewer #2: All comments have been addressed

2. Is the manuscript technically sound, and do the data support the conclusions?

Reviewer #2: (No Response)

3. Has the statistical analysis been performed appropriately and rigorously? 

Reviewer #2: (No Response)

4. Have the authors made all data underlying the findings in their manuscript fully available?

Reviewer #2: (No Response)

5. Is the manuscript presented in an intelligible fashion and written in standard English?

Reviewer #2: (No Response)

6. Review Comments to the Author

Reviewer #2: (No Response)

7. PLOS authors have the option to publish the peer review history of their article (what does this mean?). If published, this will include your full peer review and any attached files.

Reviewer #2: **Yes: **Edgar T. Walters

---

## [Author Response · Author response to Decision Letter 1]

4 Jul 2024

Thank you for the opportunity to resubmit our manuscript! We have changed Figure 3 and the body of the manuscript according to your comments.

---

## [Editor Report · Decision Letter 2]

10 Jul 2024

Depolarization of mouse DRG neurons by GABA does not translate into acute pain or hyperalgesia in healthy human volunteers

PONE-D-24-10034R2

Dear Dr. Schmelz,

We’re pleased to inform you that your manuscript has been judged scientifically suitable for publication and will be formally accepted for publication once it meets all outstanding technical requirements.

Kind regards,

Peter Wenner

Academic Editor

PLOS ONE
---

## [Editor Report · Acceptance letter]

16 Jul 2024

PONE-D-24-10034R2 

PLOS ONE

Dear Dr. Schmelz, 

I'm pleased to inform you that your manuscript has been deemed suitable for publication in PLOS ONE. Congratulations! Your manuscript is now being handed over to our production team.

Kind regards, 

on behalf of

Dr. Peter Wenner 

Academic Editor

PLOS ONE